# The Gaussian-Multinoulli Restricted Boltzmann Machine: A Potts Model Extension of the GRBM

**Nikhil Kapasi**                                                                                   *nkapasi@ucsb.edu*
*Department of Electrical and Computer Engineering*
*University of California, Santa Barbara*

**Mohamed Elfouly**                                                                                 *elfouly@ucsb.edu*
*Department of Electrical and Computer Engineering*
*University of California, Santa Barbara*

**William Whitehead**                                                                   *williamwhitehead@ucsb.edu*
*Department of Electrical and Computer Engineering*
*University of California, Santa Barbara*

**Luke Theogarajan**                                                                                  *lusthe@ucsb.edu*
*Department of Electrical and Computer Engineering*
*University of California, Santa Barbara*

**Reviewed on OpenReview:** *https://openreview.net/forum?id=3QuXwfMcKo*

## Abstract

Many real-world tasks, from associative memory to symbolic reasoning, benefit from discrete, structured representations that standard continuous latent models can struggle to express. We introduce the Gaussian–Multinoulli Restricted Boltzmann Machine (GM-RBM), a generative energy-based model that extends the Gaussian–Bernoulli RBM (GB-RBM) by replacing binary hidden units with $q$-state categorical (Potts) units, yielding a richer latent state space for multivalued concepts. We provide a self-contained derivation of the energy, conditional distributions, and learning rules, and describe the contrastive-divergence training procedure used in our recall and image-generation experiments. To separate architectural effects from parameter count, we evaluate GM-RBM under fixed visible-to-hidden weight budgets and additional hidden-size sweeps against GB-RBM baselines. On hetero-associative recall benchmarks, GM-RBM achieves competitive recall and, in several regimes, improved recall under fixed visible-to-hidden weight budgets in these experiments. The discrete $q$-ary formulation preserves standard RBM block updates. These results clarify when categorical hidden units provide a simple alternative to binary latents for discrete inference within tractable RBMs.

## 1 Introduction

Restricted Boltzmann Machines (RBMs) are energy-based models with an undirected bipartite graph structure with visible and hidden units and a subset of the broader fully connected Boltzmann machines. The lack of intralayer connections enables training tractability through parallel updates of all units within each layer (Block Gibbs updates) through a biased stochastic gradient estimator for log-likelihood (Contrastive Divergence) (Ackley et al., 1985; Smolensky, 1986; Hinton & Salakhutdinov, 2006). Although powerful, the strictly binary units in hidden and visible units make them non-ideal for dealing with multivalued data. One approach to extend their use to continuous data is to replace the visible units with Gaussian units, termed Gaussian–Bernoulli RBMs (GB-RBMs) (Hinton, 2012; Liao et al., 2022). However, their binary hidden units often struggle with inherently categorical, mutually exclusive factors. We address this mismatch by replacing each binary hidden unit with a one-of-$q$ categorical unit (Potts) (Potts, 1952; Wu, 1982; Kanter, 1988),

yielding the Gaussian–Multinoulli RBM (GM-RBM): Gaussian visibles paired with $q$-state latent hidden units. Conceptually, this keeps the RBM's simplicity but aligns inductive bias with categorical structure, effectively allowing the model to capture inherent, underlying categorical relationships. Prior work tended to retain binary latents as they fit existing tooling and samplers; in contrast, categorical slots raise practical issues (e.g., intra-slot degeneracy) and make fair comparisons challenging. To separate architectural effects from parameter count, we design comparison protocols and clarify the challenges associated with them.

Our contributions are threefold: (i) a drop-in Potts hidden layer that preserves tractable conditionals while retaining the standard RBM training pipeline; (ii) comparison protocols that separate parameter-matched and hidden-slot-matched settings; and (iii) empirical results showing that increasing $q$ can improve hetero-associative recall under fixed visible-to-hidden weight budgets with pure block Gibbs updates, while exploratory image-generation experiments suggest that GM-RBMs can produce recognizable samples under Gibbs-only sampling (Liao et al., 2022). Overall, our results suggest that replacing binary hidden units with $q$-state categorical slots can improve hetero-associative recall in higher-load regimes under fixed visible-to-hidden weight budgets, while generative results should be interpreted as proof-of-concept rather than a controlled efficiency comparison.

## 2 Background and Motivation

### 2.1 Background

Boltzmann machines (BM) are a class of energy-based models utilizing binary units ($\{-1, 1\}$) that sample from a Boltzmann probability distribution over $x = (v, h)$ given by:

$$p_\theta(x) = \frac{1}{Z(\theta)} \exp\big( - E_\theta(x) \big), \qquad Z(\theta) = \sum_x \exp\big( - E_\theta(x) \big).$$

where $\theta$ is a set of biases ($a_i$) and weights ($W_{ij}$). (Ackley et al., 1985; Smolensky, 1986; Hinton, 2002) The energy function for a BM is given by:

$$E_\theta(x) = -\sum_i a_i x_i - \tfrac{1}{2} \sum_{i \neq j} W_{ij} x_i x_j, \qquad W_{ij} = W_{ji}, \; W_{ii} = 0.$$

Log-likelihood learning decomposes into a positive (data) and negative (model) phase:

$$\frac{\partial}{\partial \theta} \log p_\theta(v) = \mathbb{E}_{p(h|v)}\Big[ - \tfrac{\partial E(v,h)}{\partial \theta} \Big] - \mathbb{E}_{p(v|h)}\Big[ - \tfrac{\partial E(v,h)}{\partial \theta} \Big],$$

which yields the classic correlation-matching rules:

$$\frac{\partial}{\partial W_{ij}} \log p_\theta(v) = \langle x_i x_j \rangle_{\text{data}} - \langle x_i x_j \rangle_{\text{model}}, \qquad \frac{\partial}{\partial a_i} \log p_\theta(v) = \langle x_i \rangle_{\text{data}} - \langle x_i \rangle_{\text{model}}.$$

However, correlation-matching rules are intractable for most real-world data (Liao et al., 2022). One way to overcome this is to use the Restricted Boltzmann machine: Restricted Boltzmann machines split the graph into a bipartite structure with visible units $v$ and hidden units $h$, removing within-layer edges so that the joint probability distribution can be broken down into simpler conditional probabilities enabling block Gibbs sampling. For binary visibles $v \in \{0, 1\}^n$ and binary hiddens $h \in \{0, 1\}^m$,

$$E(v, h) = -b^\top v - c^\top h - v^\top W h,$$

$$p(h_j = \mathbf{1} \mid v) = \sigma\big( (W^\top v)_j + c_j \big), \qquad p(v_i = \mathbf{1} \mid h) = \sigma\big( (W h)_i + b_i \big),$$

and the learning updates specialize to moment matching between the two layers,

$$\Delta W \propto \mathbb{E}[v h^\top]_{\text{data}} - \mathbb{E}[v h^\top]_{\text{model}}, \quad \Delta b \propto \mathbb{E}[v]_{\text{data}} - \mathbb{E}[v]_{\text{model}}, \quad \Delta c \propto \mathbb{E}[h]_{\text{data}} - \mathbb{E}[h]_{\text{model}}.$$

Contrastive divergence (CD) approximates model expectations by alternating $h \sim p(h \mid v)$ and $v \sim p(v \mid h)$ for $k$ steps of data CD-$k$ or for a persistence chain (persistent CD) (Hinton, 2002; Tieleman, 2008; Hinton, 2012). Usually, both hidden and visible units are taken as binary, but for continuous data it is natural to use Gaussian visible units (Hinton, 2012; Liao et al., 2022). In the Gaussian–Bernoulli RBM, visibles are Gaussian with diagonal variance $\sigma^2$ and hiddens remain binary. A convenient parameterization is:

$$E(v, h) = \sum_i \frac{(v_i - \mu_i)^2}{2\sigma_i^2} - \sum_{i,j} \frac{v_i}{\sigma_i^2} W_{ij} h_j - \sum_j b_j h_j,$$

which gives closed-form conditionals used inside CD,

$$p(v \mid h) = \mathcal{N}\big(\mu + Wh,\ \mathrm{diag}(\sigma^2)\big), \qquad p(h_j = 1 \mid v) = \mathrm{Sigmoid}\Big(\big[W^\top (v \oslash \sigma^2)\big]_j + b_j\Big).$$

The corresponding stochastic-gradient updates keep the same positive/negative structure but respect the visible scaling,

$$\Delta \boldsymbol{W} \propto \left\langle \left(\frac{\mathbf{v}}{\boldsymbol{\sigma}^2}\right) \mathbf{h}^\top \right\rangle_{\mathrm{data}} - \left\langle \left(\frac{\mathbf{v}}{\boldsymbol{\sigma}^2}\right) \mathbf{h}^\top \right\rangle_{\mathrm{model}},$$

$$\Delta \boldsymbol{\mu} \propto \left\langle \frac{\mathbf{v} - \boldsymbol{\mu}}{\boldsymbol{\sigma}^2} \right\rangle_{\mathrm{data}} - \left\langle \frac{\mathbf{v} - \boldsymbol{\mu}}{\boldsymbol{\sigma}^2} \right\rangle_{\mathrm{model}}$$

$$\Delta \boldsymbol{b} \propto \langle \boldsymbol{h} \rangle_{\mathrm{data}} - \langle \boldsymbol{h} \rangle_{\mathrm{model}}$$

In this work we extend the role of the binary hidden variables by replacing each Bernoulli with a one-of-$q$ categorical (Potts) slot, aligning the latent prior with mutually exclusive factors while preserving the RBM's block-sampling tractability and the Gaussian visible layer.

## 2.2 Motivating Principles

Many perceptual and symbolic factors are naturally *categorical and mutually exclusive.* Approximating such structure with many Bernoulli latents (as in a GB-RBM) forces variants to be represented by co–activating subsets of units, which encodes information across the hidden layer and yields ambiguous codes.

The Gaussian–Multinoulli RBM (GM-RBM) encodes each factor as a *one-of-$q$* slot. With $m$ slots and hidden configuration $h = (h_1, \ldots, h_m)$ where $h_j \in \{1, \ldots, q\}$, the visible conditional remains Gaussian,

$$p(v \mid h) = \mathcal{N}\Big(b + \sum_{j=1}^m W_{:,j}^{(h_j)},\ I\Big),$$

so each chosen state contributes a single template vector and the mean is a sum of selected templates. This preserves the RBM's locally linear structure encoding the continuous variables using the set of latent discrete variables. Each categorical state within a slot has its own template vector, so changing the selected state changes the contribution of that slot without requiring coordinated activation across multiple binary units. In practice, this gives each hidden slot more categorical degrees of freedom than simply enforcing a one-hot encoding across multiple Bernoullis, which intrinsically lack intra-layer coupling.

GM-RBM is a modification where we replace binary hidden units with categorical slots but retain Gaussian visibles and standard RBM conditionals. We assess GM-RBM on hetero-associative recall and image modeling. To distinguish architectural effects from parameter-count effects, we use both *parameter-matched (fixed visible-to-hidden weight budget)* and *hidden-slot-matched (fixed hidden-unit count)* comparisons, which appear later.

## 3 Theoretical Foundations of the Gaussian–Multinoulli RBM (GM-RBM)

The Gaussian–Multinoulli RBM (GM-RBM) replaces binary hidden units with discrete $q$-state categorical variables while keeping a continuous visible layer. This yields a combinatorial latent space with simple, closed-form conditionals.

### 3.1 Notation

Let $v \in \mathbb{R}^n$ be the visible vector. The hidden code is $h = (h_1, \ldots, h_m)$ with $h_j \in \{1, \ldots, q\}$. Parameters are: visible bias $b \in \mathbb{R}^n$; hidden bias $c_{j,k} \in \mathbb{R}^m$; and state-specific templates $W_{:,j}^{(k)} \in \mathbb{R}^n$. Define the conditional mean

$$\mu(h) = b + \sum_{j=1}^m W_{:,j}^{(h_j)}$$

### 3.2 Energy, Joint, and Conditionals

The energy is

$$E(v, h) = \tfrac{1}{2} \sum_{i=1}^n (v_i - b_i)^2 - \sum_{j=1}^m c_{j,h_j} - \sum_{i=1}^n \sum_{j=1}^m W_{ij}^{(h_j)} v_i.$$

Completing the square gives $E(v, h) = \frac{1}{2} \|v - \mu(h)\|_2^2 + K(h)$ with $K(h) = \frac{1}{2}(\|b\|_2^2 - \|\mu(h)\|_2^2) - \sum_j c_{j,h_j}$. The joint is $p(v, h) \propto \exp(-E(v, h))$. The conditionals are

$$p(v \mid h) = \mathcal{N}\big(\mu(h), \mathbf{1}\big)$$

$$p(h_j = k \mid v) = \frac{\exp\big(c_{j,k} + (W_{:,j}^{(k)})^\top v\big)}{\sum_{k'=1}^q \exp\big(c_{j,k'} + (W_{:,j}^{(k')})^\top v\big)} = \mathrm{Softmax}(c_{j,k'} + (W_{:,j}^{(k')})^\top v)$$

### 3.3 Architecture and Special Cases

Each slot contributes one of $q$ templates, so the codebook $\{\mu(h) : h \in \{1, \ldots, q\}^m\}$ has size $q^m$. When $q = 2$ and parameters are tied as $W_{:,j}^{(1)} - W_{:,j}^{(2)} = \widetilde{W}_{:,j}$ and $c_{j,1} - c_{j,2} = \widetilde{b}_j$ with a corresponding recentering of $b$, the GM-RBM reduces to a Gaussian–Bernoulli RBM. There is no requirement that $q$ be even; experiments may vary $q$ according to task. It is also possible to use usual variance reweighting as opposed to unit variance used in the GB-RBM.

### 3.4 Prior work on categorical-unit RBMs

Multinomial/softmax $q$-state units in RBMs have appeared in prior work and we differentiate our work from this literature as enumerated below.

1. **Welling et al. (2005)** introduced *Exponential Family Harmoniums* (EFHs), which generalize RBM units from Bernoulli to any exponential family member—including multinomial/categorical units as a special case. The GM-RBM can be viewed as an EFH with Gaussian visible units and multinomial hidden units. However, Welling et al. focused on the general framework and its application to information retrieval with Poisson visibles; they did not explore the specific Gaussian–multinoulli pairing, fair comparison protocols, or the empirical consequences for associative memory and mixing.

2. **Montúfar & Morton (2015)** provided a rigorous theoretical treatment of *discrete RBMs* (also called multinomial or softmax RBMs), where each unit has a finite state space $\{0, 1, \ldots, r-1\}$. Their analysis addresses representational capacity bounds and dimension computation for these models. Our work complements their theoretical results with an empirical demonstration that the Gaussian–Potts combination yields concrete performance gains on structured memory tasks under controlled comparison protocols. While they established that discrete RBMs with multi-state hidden units have greater representational capacity than binary RBMs, the mixing properties of these models under block Gibbs sampling have not been studied in this setting. Our ESS diagnostics provide

empirical evidence, in this benchmark, that Potts hidden units can improve mixing under block Gibbs sampling.

3. **Tran et al. (2011)** introduced *Mixed-Variate RBMs* capable of modeling categorical, multicategorical, ordinal, and continuous variables simultaneously. However, the categorical units in their framework appear on the *visible* side to handle mixed-type input data, while the hidden layer remains binary. In contrast, our GM-RBM places categorical units on the *hidden* side to enrich the latent representation while keeping Gaussian visibles.

4. **Salakhutdinov & Hinton (2009b)** introduced the *Replicated Softmax* model, which uses softmax visible units for topic modeling of word-count data. Again, the multinomial structure is on the visible side with binary hidden units, whereas our contribution places it on the hidden side.

What distinguishes our work is not the Potts unit concept itself, but: (a) the specific Gaussian-visible / Potts-hidden pairing and its closed-form conditionals; (b) the parameter-matched and hidden-slot-matched comparison protocols that separate architectural effects from parameter count; and (c) the empirical demonstration that this combination yields statistically significant gains on hetero-associative memory benchmarks with pure Gibbs sampling (no Langevin).

### 3.5 Parameter Count and Fair Comparison Protocols

GM-RBM has $n$ parameters for $b$, plus $mq$ for $c$, plus $nmq$ for $W$, for a total of $n + mq\,(1+n)$. A GB-RBM with $m'$ binaries has $n + m' + nm'$. We evaluate under two protocols:

- **Parameter-matched:** The visible-to-hidden weight budget is held fixed at $n_w$ (approximately 800k weights in our main recall experiments). For a model with $q$ Potts states, the number of hidden units is set as $H = \lfloor n_w/(n_v \cdot q) \rfloor$, so that $n_v \cdot H \cdot q \approx n_w$ for all $q$.

- **Hidden-slot-matched:** The number of hidden units $H$ is held fixed across models regardless of $q$, so higher-$q$ models have proportionally more parameters and a larger latent assignment space. This setting is not parameter matched; it tests how performance changes when each hidden slot is allowed more categorical states.

These regimes separate fixed-parameter comparisons from sweeps that expand the per-slot categorical state space.

### 3.6 Learning and Negative-phase Sampling

The log-likelihood gradient satisfies

$$\frac{\partial}{\partial \theta} \log p(v) = \mathbb{E}_{p(h|v)}\left[-\frac{\partial E(v,h)}{\partial \theta}\right] - \mathbb{E}_{p(v|h)}\left[-\frac{\partial E(v,h)}{\partial \theta}\right], \quad \theta \in \{b,c,W\}.$$

We approximate the model expectation with short Markov chains.

**Block Gibbs.** Alternate $h \sim p(h \mid v)$ using the per-slot softmax and $v \sim p(v \mid h) = \mathcal{N}(\mu(h), \mathbf{1})$. The visible draw is exact and parameter free.

**Gibbs with visible Langevin.** Some implementations replace the exact Gaussian draw with an unadjusted Langevin step using $\nabla_v \log p(v \mid h) = \mu(h) - v$ (Liao et al., 2022):

$$v_{t+1} = v_t + \tfrac{\varepsilon^2}{2}\big(\mu(h_t) - v_t\big) + \varepsilon\,\xi_t, \quad \xi_t \sim \mathcal{N}(0, \mathbf{1}), \ \varepsilon > 0.$$

This introduces a stepsize–dependent discretization error; for fixed $h_t$, multiple Langevin steps only approximate the exact conditional. In many Gaussian RBM variants, each visible dimension is allowed its own

variance, so that $p(v \mid h) = \mathcal{N}(\mu(h), \Sigma)$ with a diagonal covariance $\Sigma = \text{diag}(\sigma^2)$. In this case, the gradient becomes $\Sigma^{-1}(\mu(h_t) - v_t)$, and the drift term is modified to $(\mu(h_t) - v_t)/\sigma^2$ (applied elementwise). Intuitively, inserting a visible Langevin move between hidden-layer updates allows hidden units to exchange information indirectly through the visible layer and can encourage more cooperative encodings.

In contrast to Langevin-based visible updates, we use exact block Gibbs updates for the GM-RBM. The visible Langevin step is an approximate sampler for the same conditional distribution $p(v \mid h)$ and introduces a stepsize-dependent approximation. In our experiments, we therefore report whether a visible Langevin step is used, the stepsize, the number of steps, and whether chains are persistent.

**Sampler cost, mixing, and our choice.** The visible Langevin variant adds at least one extra update and a stepsize hyperparameter per negative step relative to an exact Gaussian draw. In our setting the Gibbs visible update samples exactly from $N(\mu(h), \mathbf{1})$ with a single noise vector and no stepsize tuning. We report ESS diagnostics in Appendix A (Table 1); in this benchmark, the binary baseline reaches 82.9% ESS over 500 Gibbs steps while $q \geq 4$ models reach the maximum measured ESS of 100% with comparable final energies.

### 3.7 Key Properties

- Locally linear: given $h$, $p(v \mid h)$ is Gaussian with fixed covariance.

- Globally discrete: the means form a finite codebook indexed by discrete slots.

- Modular: slots contribute additively in $\mu(h)$ and independently in $p(h \mid v)$.

## 4 Hetero-associative Memory

Hetero-associative memory refers to a system's capability to learn paired associations between distinct patterns, allowing the retrieval of a target pattern (e.g., a response word) when presented with a corresponding stimulus (e.g., a cue word) (Hopfield, 1982; Kosko, 1988; Morales & Pineda, 2024). This concept, rooted in cognitive modeling and neural computation, was further explored by Hinton in 1981 Anderson & Hinton (1981) and later employed in language-related tasks using Gaussian-Bernoulli Restricted Boltzmann Machines (GB-RBMs) (Tsutsui & Hagiwara, 2019). However, binary hidden units can limit how such multivalued associations are represented.

In the higher-$q$ regimes, GM-RBMs deliver consistently higher recall than the GB-RBM baseline, with gains becoming most pronounced at larger associative-memory loads. This comparison should be read with the sampler difference in mind: the GM-RBM uses Gibbs sampling, whereas the GB-RBM baseline uses hybrid Gibbs–Langevin sampling.

### 4.1 Experimental Setup

We replicated the experimental setup in Tsutsui and Hagiwara, constructing a word-pair dataset representing conceptual relationships (e.g., "apple is-a fruit") (Tsutsui & Hagiwara, 2019). We randomly selected pairs from WordNet (Princeton University (2010)), excluding compound or incomplete entries and sampled 500-3,000 word pairs to create smaller datasets for training and testing scalability.

Each word pair is treated as a directional association task and encoded as a concatenated embedding vector (Mikolov et al., 2013), mirroring the hetero-associative memory objective of recalling a target concept from a given stimulus.

We trained a 200-dimensional Continuous Bag of Words (CBOW) Word2Vec model (Mikolov et al., 2013) on the word-pair dataset (100 iterations, window size 5, no frequency cutoff). We normalized each vector dimension to zero mean and unit variance to mitigate small-magnitude issues. For each stimulus-response pair, embeddings were concatenated to form a 400-dimensional input vector for the RBM's visible layer.

We used a compute node (Intel Xeon Gold 6154 ×2, 512 GB RAM, NVIDIA Tesla P40) running Red Hat Enterprise Linux 8. Models were trained with CUDA-accelerated PyTorch (v1.13) on a single GPU.

Experiments were automated via a modular framework for data loading, configuration, visualization, and checkpointing.

## 4.2 Training

Our GM-RBM learned to associate stimulus–response word embeddings. Datasets comprised semantically related word pairs (e.g., doctor–nurse, sun–light). Following a similar procedure to Tsutsui and Hagiwara Tsutsui & Hagiwara (2019), we trained a CBOW Word2Vec model (Řehůřek & Sojka, 2010) on these two-word sentences to capture in-domain semantics, producing 200-dimensional embeddings (100 iterations, window size 5, no frequency cutoff).

We normalized each embedding dimension to zero mean and unit variance to avoid numerical instability. The stimulus and response embeddings were concatenated into a 400-dimensional visible layer vector.

We trained the GB-RBM using contrastive divergence with a two-step Gibbs burn-in and Gibbs–Langevin sampling. In contrast, the GM-RBM variant relied solely on standard Gibbs sampling. Notably, in the special case of $q = 2$, the GM-RBM formulation becomes almost equivalent to the GB-RBM: the only difference is that in the GB-RBM, the two weight matrices are essentially negatives of one another. Hidden-unit counts were scaled inversely with the number of Potts states to keep the visible-to-hidden weight budget approximately constant. Training used Adam ($LR = 10^{-4}$) with mini-batches of 64. We evaluated recall by inferring responses via Gibbs sampling and selecting the nearest neighbor; accuracy was the percentage of correct matches.

We stopped training early when recall accuracy reached 0.98 on the validation set, when the standard deviation of validation accuracy over 20 checkpoints fell below 0.01, or when no improvement was observed for 10 consecutive checkpoints.

## 4.3 Results

To isolate the effect of the Potts hidden units, we performed two key experiments. The first was a parameter-matched comparison where the visible-to-hidden weight budget was kept approximately constant while $q$ was increased. The second was a hidden-slot-matched sweep where $q$ was held constant while the number of hidden units was increased. Both sweeps were performed across varying dataset sizes, measuring recall accuracy for pairs embedded in Word2Vec (Mikolov et al., 2013).

Each marker in Figures 2 and 3 denotes an independently trained model using identical early-stopping criteria. Figure 1 reports mean ± standard deviation across 10 independent seeds, showing that the observed gains are stable across seeds.

For data shown in Figure 2, the visible-to-hidden weight budget was held approximately constant; hidden-layer sizes were decreased proportionally to the cardinality of the Potts state $q$ (see Table 1).

It is also important to note that $q = 2$ for the GM-RBM is a Potts-node case where weights do not have to be constrained as negatives of one another. The original GB-RBM has higher recall than the GM-RBM for $q = 2$, possibly due to this difference. In the higher-$q$ regimes ($q = 4, 6, 8, 10$), the GM-RBM models achieve higher recall than the GB-RBM baseline on this benchmark, while using Gibbs updates rather than the Gibbs–Langevin update used by the baseline (Liao et al., 2022).

Because a fixed visible-to-hidden weight budget may obscure how hidden-layer width affects performance, we also swept the number of hidden nodes and dataset sizes as shown in Figure 3. In this hidden-slot-matched sweep, GM-RBM $q = 2$ and the GB-RBM both degrade when $N > 2000$, while the GM-RBM $q = 4$ maintains higher retrieval accuracy.

### 4.3.1 Parameter matched $q$ sweep

To show the direct effect of the additional Potts states, we held the visible-to-hidden weight budget approximately constant while varying the number of Potts states $q$, isolating the effect of the state space's structure on hetero-associative performance. Figure 2 plots retrieval accuracy against the number of asso-

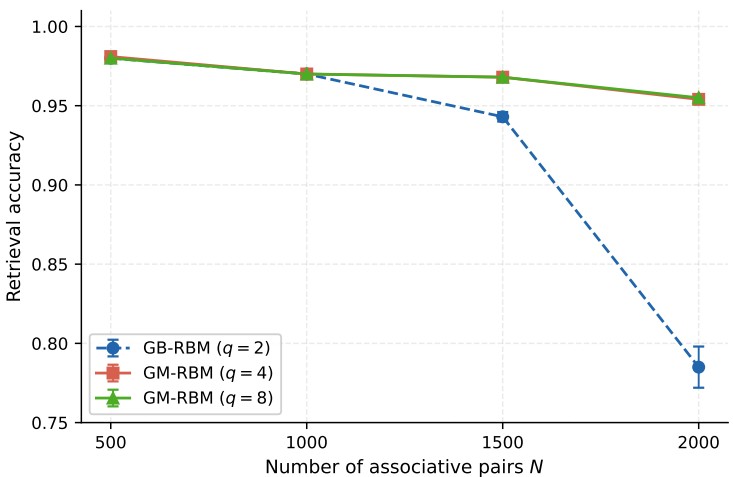

Figure 1: Retrieval accuracy on the hetero-associative memory task, reported as mean $\pm$ standard deviation across 10 independent seeds. The visible-to-hidden weight budget is fixed at approximately 800k weights. The advantage of Potts hidden states is most pronounced in the high-load regime at $N = 2000$, where the GB-RBM drops to 78.5% while GM-RBM ($q \geq 4$) maintains $\sim$95.4%.

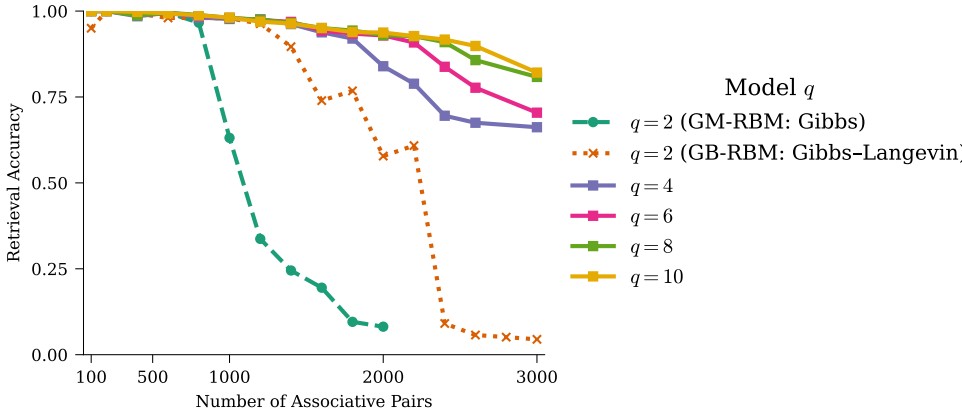

Figure 2: Retrieval accuracy versus the number of associative pairs for different numbers of Potts states (q) in a parameter-matched GB-RBM and GM-RBM setup

ciative pairs in the training set for each $q$. For the binary case $q = 2$, the GB-RBM with Gibbs–Langevin sampling maintains near-perfect accuracy at small dataset sizes but collapses sharply beyond 1000 pairs, whereas the GM-RBM using only Gibbs sampling degrades more rapidly. In contrast, models with higher state cardinality ($q = 4, 6, 8, 10$) sustain almost perfect retrieval up to roughly 1200–1500 pairs and exhibit a more gradual decline as a function of dataset size $N$. Higher-$q$ GM-RBM variants maintain higher recall than the binary baselines at larger $N$, though the more controlled 10-seed ablation in Table 2 suggests that differences among $q \geq 4$ are small under the fixed visible-to-hidden weight-budget $N = 2000$ setting.

To keep the visible-to-hidden weight budget approximately constant across different Potts-state configurations, we solve for the number of hidden units $n_h$ by dividing the total budget of weight parameters $n_w$ by the number of Potts states $q$. Since each hidden unit with $q$ Potts states contributes $q$ weight vectors of length equal to the number of visible units $n_v$, the number of hidden units is given by

$$n_h = \left\lceil \frac{n_w}{n_v \cdot q} \right\rceil \tag{1}$$

We chose $n_w = 800{,}000$, ensuring that the binary case $q = 2$ matched the optimal hidden-unit count of $1{,}000$ reported by Tsutsui and Hagiwara Tsutsui & Hagiwara (2019) for the GB-RBM. This ensures that, as we increase the number of Potts states $q$, we reduce the number of hidden units proportionally so that every model has approximately the same visible-to-hidden weight budget.

Table 1: Potts-state $q$ versus number of hidden units

|  | $q = 2$ | $q = 4$ | $q = 6$ | $q = 8$ | $q = 10$ |
|---|---|---|---|---|---|
| Hidden Units | 1000 | 500 | 333 | 250 | 200 |

These results suggest that increasing the number of discrete states can improve memory robustness under a fixed visible-to-hidden weight budget. The effect is most pronounced at high load ($N = 2000$), where the categorical advantage is large and stable across seeds (see Figure 1).

### 4.3.2 Hidden nodes sweep

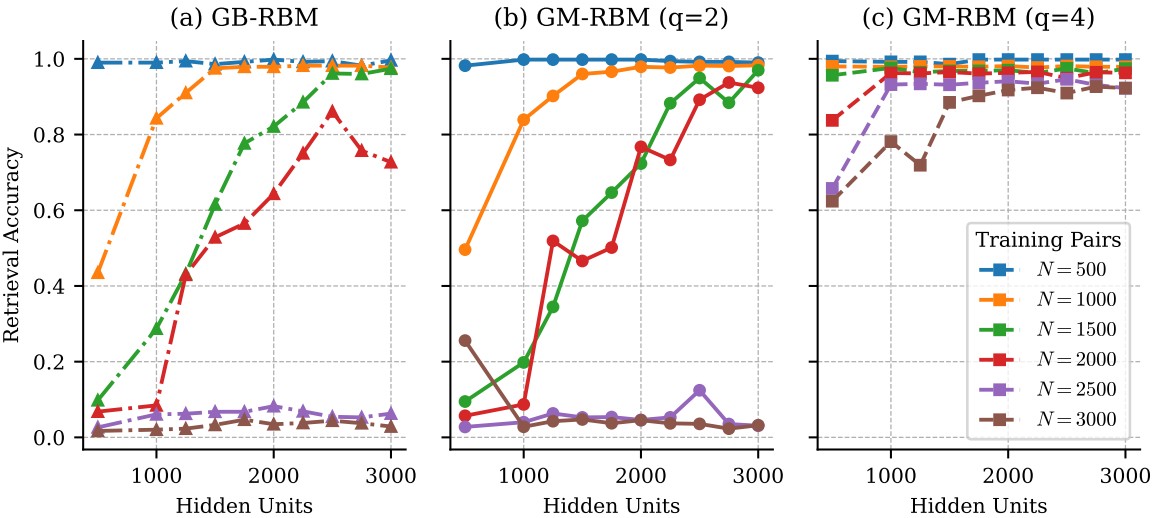

Figure 3: Retrieval accuracy as a function of *hidden-unit count* (not parameter-matched) for different model variants and dataset sizes. The number of hidden units is varied from 500 to 3000 with $q$ held fixed. (a) GB-RBM (Gibbs–Langevin update), (b) GM-RBM with $q = 2$ (Gibbs update), (c) GM-RBM with $q = 4$ (Gibbs update). Each curve corresponds to a different number of training word-pair examples ($N \in \{500, 1000, 1500, 2000, 2500, 3000\}$).

To show how varying the hidden-layer dimensionality affects retrieval performance across dataset sizes, we swept the number of hidden units from 500 to 3000 while fixing the state cardinality $q$. Figure 3 presents retrieval accuracy curves for three models: a standard Gaussian–Bernoulli RBM (GB-RBM), a GM-RBM with $q = 2$, and a GM-RBM with $q = 4$. Each curve represents a different dataset size ($N = 500, 1000, 1500, 2000, 2500, 3000$ associative word pairs).

For the binary GM-RBM ($q = 2$), accuracy is near-perfect at small loads but degrades sharply as $N$ increases, recovering only when hidden units exceed 1500. In contrast, the Potts-based GM-RBM with $q = 4$ maintains over 90% accuracy across all dataset sizes with just 1000 hidden units. The GB-RBM baseline requires roughly 2500 hidden units to achieve similar performance at large scales (e.g., $N = 2500$).

These findings suggest a trade-off between hidden-layer dimensionality and state complexity in this benchmark: increasing $q$ can reduce the hidden-unit requirement for robust associative recall.

### 4.3.3 Ablation over $q$ under fixed visible-to-hidden weight budget

To characterize when increasing $q$ stops helping, we extended the sweep to $q \in \{2, 4, 6, 8, 10, 12, 16, 20\}$ at fixed $N = 2000$ and the 800k visible-to-hidden weight budget, running 10 independent seeds per configuration (Table 2). The binary baseline ($q = 2$, $H = 1000$) reaches only 80.2% accuracy. All Potts configurations ($q \geq 4$) plateau at $\approx 95.6\%$ with variance below $\sigma = 0.002$, even at $q = 20$ with only 100 hidden units. This suggests that the primary gain in this benchmark comes from the binary-to-categorical transition, rather than from fine-tuning $q$ beyond 4.

Table 2: Retrieval accuracy under fixed 800k visible-to-hidden weight budget ($N = 2000$, 10 seeds).

| $q$ | $H$ | Accuracy (mean $\pm$ std) | Time (s/epoch) |
|-----|-----|---------------------------|----------------|
| 2 | 1000 | $0.802 \pm 0.007$ | 1.2 |
| 4 | 500 | $0.956 \pm 0.001$ | 1.4 |
| 6 | 333 | $0.956 \pm 0.001$ | 1.5 |
| 8 | 250 | $0.956 \pm 0.001$ | 1.6 |
| 10 | 200 | $0.957 \pm 0.001$ | 1.7 |
| 12 | 166 | $0.956 \pm 0.001$ | 1.8 |
| 16 | 125 | $0.956 \pm 0.001$ | 2.1 |
| 20 | 100 | $0.956 \pm 0.001$ | 2.4 |

## 5 Auto-associative memory

As a proof of concept for the generative capability of our GM-RBM, we performed a replication of the generative experiments from the original GB-RBM paper Liao et al. (2022). On MNIST, we trained for 500 epochs, and on CelebA for 100 epochs. Starting from i.i.d. Gaussian noise in the visible layer and running 1,000 steps of Gibbs sampling, we obtained samples (see Figure 4) that qualitatively demonstrate the GM-RBM's ability to produce recognizable generative outputs.

### 5.1 Experimental setup

We use two datasets for exploratory image-generation experiments: **MNIST** (28×28 grayscale handwritten digits) (LeCun et al., 1998) and **CelebA** (center-cropped and resized RGB facial images at 64×64 resolution) (Liu et al., 2015).

All image datasets are normalized to zero mean and unit variance per channel (Liao et al., 2022). CelebA images are center-cropped to 140×140 before downsampling to 64×64 (Liu et al., 2015). For each dataset, we generate training and evaluation splits using standard protocols.

Experiments were performed on a dedicated compute node equipped with dual Intel Xeon Gold 5218 CPUs, 512 GB of RAM, and eight NVIDIA RTX 6000 GPUs connected via NVLink. All models were trained using CUDA-accelerated PyTorch (v1.13+) with single-GPU execution unless otherwise specified.

### 5.2 Results

We observe that the $q = 4$ GM-RBM generates visually identifiable face/digit samples under standard Gibbs sampling (see Figure 4a and 4b; Table 3). These results motivate further controlled comparisons under matched hyperparameter budgets (see Section 6.1).

Because these runs use different hidden sizes, training schedules, and sampling procedures, the image-generation results should be interpreted as exploratory proof-of-concept evidence rather than a controlled head-to-head comparison. They show that GM-RBMs can produce recognizable samples under Gibbs sampling, but they do not by themselves establish superior training efficiency or generative performance.

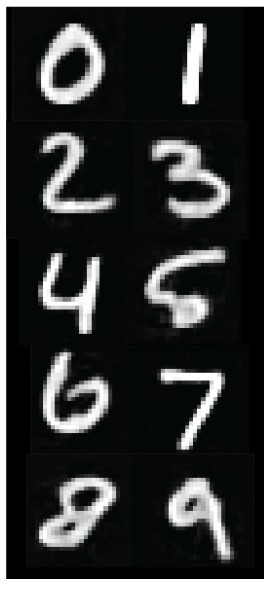

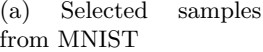

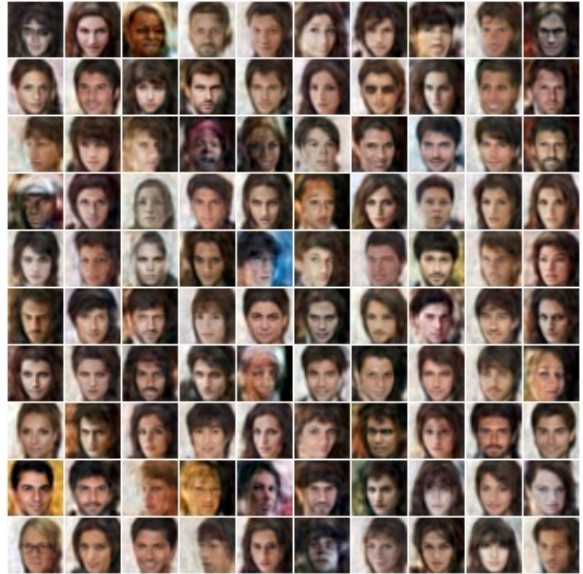

(a) Selected samples from MNIST

(b) Generation of CelebA images from random noise with $q = 4$

Figure 4: Sampled results from GM-RBM of MNIST and CelebA datasets.

Table 3: Hyperparameter settings for GB-RBM and GM-RBM on MNIST and CelebA

| Hyperparameter | MNIST | | CelebA | |
|---|---|---|---|---|
| | GB-RBM | GM-RBM | GB-RBM | GM-RBM |
| Number of states | 2 | 4 | 2 | 4 |
| Sampling style | Gibbs–Langevin | Gibbs | Gibbs–Langevin | Gibbs |
| Hidden nodes | 4096 | 2048 | 10000 | 5000 |
| Visible nodes | 784 | 784 | 3072 | 3072 |
| Epochs trained | 3000 | 500 | 10000 | 100 |

### 5.2.1 Quantitative sample quality (FID)

Table 4: Exploratory FID (↓) comparison; training schedules and samplers are not matched. Best overall in **bold**.

| Model | Potts States | FID |
|---|---|---|
| GM-RBM | $q = 2$ | 67.08 |
| GM-RBM | $q = 4$ | 56.09 |
| GM-RBM | $q = 6$ | **53.07** |
| GB-RBM | N/A | 60.06 |

To complement the qualitative samples, we report Fréchet Inception Distance (Heusel et al., 2018) (FID; ↓ is better) for an exploratory comparison in Table 4. Because the training schedules, hidden sizes, and samplers differ, these FID values should not be interpreted as a controlled head-to-head efficiency comparison or attributed solely to architecture.

In this exploratory setting, GM-RBM ($q$=6) obtains a lower FID than the GB-RBM baseline (53.07 vs 60.06). However, because the budgets are not fully matched, this result should be viewed as evidence that GM-RBMs can generate recognizable samples under Gibbs-only updates, not as a definitive comparison of generative performance. It is also important to note that the CelebA dataset does not necessarily have clear relational qualities, so these scores may reflect interactions among architecture, sampler, and training schedule rather than the Potts representation alone.

## 6 Limitations and future work

While the Gaussian–Multinoulli RBM (GM-RBM) shows improved recall on our hetero-associative benchmark under fixed visible-to-hidden weight budgets, important limitations and next steps remain.

### 6.1 Limitations of the current model

**1. Sampling constraints and rationale.** Our GM-RBM uses *pure block Gibbs* (exact Gaussian visible draw + per-slot softmax posteriors). We intentionally avoid visible-space Langevin during training because it adds step-size hyperparameters, extra updates, and discretization error, increasing compute relative to a single exact Gaussian draw.

**2. Evaluation scope and scaling.** Most of our empirical gains are on hetero-associative recall with in-domain Word2Vec embeddings and proof-of-concept image generation. This leaves open: (a) robustness to larger or off-the-shelf embeddings (e.g., large CBOW/skip-gram trained on broader corpora), (b) transfer to other modalities (audio, trajectories, time series), and (c) deeper stacks (e.g., GM front-ends for DBMs/DBNs). We also observe retrieval degradation as $N$ grows in some settings; diagnosing whether this arises from chain mixing, parameter-matching choices, or dataset effects will require ablations (burn-in length, persistence, $q$ vs. $m$, negative-phase steps).

**3. Training budget and stability.** Our generative experiments used short schedules (e.g., 500 epochs on MNIST, 100 on CelebA), so they should be treated as exploratory. Longer training with early-stopping and variance monitoring could reveal stability regimes, impacts on sample diversity, and when different negative-phase samplers are beneficial.

### 6.2 Broader impact

#### 6.2.1 Potts units in energy transformers

Energy Transformers minimize a global energy via recurrent updates. Replacing binary hidden units with $q$-state Potts slots increases the latent assignment space from $2^H$ to $q^H$ and may affect attractor overlap:

$$E_{\text{mem}}(h) \; = \; -\sum_{\mu=1}^{M}\sum_{i=1}^{H} \delta(h_i, \xi_i^{\mu}). \tag{2}$$

We hypothesize potential selectivity and robustness changes under higher $q$ and are empirically exploring GM-RBM components as ET memories; careful comparisons should use either hidden-slot-matched or parameter-matched protocols as above. (Schröder et al., 2024; Hoover et al., 2023)

As a front-end to DBMs, a Gaussian–Potts encoder maps $\mathbb{R}^D$ into a $q^H$ latent space while preserving layer-wise Gibbs training. Future work should examine stability under deeper stacks and compare against increased Bernoulli width under parameter-matched or hidden-slot-matched protocols. (Hinton et al., 2006; Salakhutdinov & Hinton, 2009a; Morales & Pineda, 2024; Jang et al., 2017; Maddison et al., 2017; Oh et al., 2022)

Many foundational generative families, RBMs, DBNs/DBMs, Hopfield-style memories, VAEs with Bernoulli latents, Gumbel-Softmax relaxations, autoregressive binarized pixel models, BNNs, SNNs, and discrete diffusion—lean on binary or categorical sampling (Hinton et al., 2006; Salakhutdinov & Hinton, 2009a; Hopfield, 1982; Kosko, 1988; Kingma & Welling, 2014; Jang et al., 2017; Maddison et al., 2017; van den Oord et al.,

2016a;b; Hubara et al., 2016; Neftci et al., 2019; Ho et al., 2020). Swapping binary units for Potts slots increases representational granularity and may reduce interference among stored patterns under appropriate matching protocols. A broad, hidden-slot- or parameter-matched survey across these families is an open, high-impact direction.

Binary and Potts one-hot codes map naturally to LUTs and bitwise logic. Sparse/event-driven readout and lightweight on-chip softmax enable efficient FPGA/ASIC/neuromorphic realizations; exploring SPAD-assisted annealing or mixed-signal implementations for categorical slots is promising future work (Whitehead et al., 2023).

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
