# OpenReview forum: "The Gaussian-Multinoulli Restricted Boltzmann Machine: A Potts Model Extension of the GRBM"
_TMLR — Decision pending for TMLR_

### Review · Reviewer_Fko1 · 2026-01-05

**Summary Of Contributions:**

This paper contributes a relatively clean and well-motivated extension of the Gaussian–Bernoulli RBM by replacing binary hidden units with q-state categorical (Potts) units, resulting in the proposed Gaussian–Multinoulli RBM. The main contribution is not just the model itself, but the careful framing of why categorical slots better match mutually exclusive latent structure and how this can be evaluated fairly against binary baselines. The authors provide a clear derivation of the energy function and conditionals, show how the model reduces to a GB-RBM in a special case, and place a lot of emphasis on capacity- and parameter-matched comparisons. Empirically, the strongest contribution is the hetero-associative memory results, where GM-RBMs consistently outperform GB-RBMs under matched budgets, even when trained with simpler sampling. The generative experiments on MNIST and CelebA, along with FID scores, serve more as supporting evidence that the approach is viable beyond memory tasks, rather than as a definitive claim of state-of-the-art generative performance .

**Audience:**

Yes

**Audience Explanation:**

Yes, I think clearly so. Researchers interested in energy-based models, discrete latent variables, associative memory, or the revival of RBM-style architectures would likely find this paper relevant. The work sits in a somewhat niche but active space: people who care about alternatives to large autoregressive or diffusion models, or about structured and interpretable latent representations, are a natural audience. The paper also connects to recent interest in Potts models, energy transformers, and discrete inference, which helps anchor it in ongoing conversations rather than purely historical RBM work.

**Broader Impact Concerns:**

The broader impact concerns here are relatively mild. The model itself does not introduce obvious societal risks beyond those common to generative models trained on image or text data. The CelebA experiments inherit the usual concerns around facial datasets, but these are standard and not amplified by the proposed method. On the positive side, the work could encourage more parameter-efficient and interpretable models, which is generally beneficial. If Potts-style latent units enable better memory capacity or robustness, there may be downstream applications in low-resource or hardware-constrained settings, which is arguably a positive impact. Overall, there are no clear red flags, but also no immediate high-stakes societal implications; the impact is primarily methodological and scientific rather than directly social.

**Claims And Evidence:**

Yes

**Claims Explanation:**

Overall, the core claims are mostly supported, though not uniformly so. The paper’s main technical claims—that replacing Bernoulli hidden units with Potts-style categorical units yields better capacity usage, improved hetero-associative recall, and competitive or improved generative performance under fair matching—are backed by reasonably clear derivations and a fairly careful experimental design. The theoretical sections are self-contained and consistent, and the reduction to the GB-RBM in the q=2 case is clearly explained, which helps ground the novelty. Empirically, the hetero-associative memory experiments are the strongest evidence: the parameter-matched and hidden-unit sweeps are well motivated, and the trends in Figures 1 and 2 are large enough that the qualitative conclusions seem robust. However, some claims are presented a bit more confidently than the evidence fully warrants. For example, statements about “fast mixing” and reduced need for Langevin updates are plausible and partially supported, but the diagnostics are not very deep, and statistical variability is largely dismissed rather than quantified. Similarly, the generative image results rely partly on visual inspection and limited FID reporting, which is suggestive but not fully convincing as a general claim of superiority.

**Requested Changes:**

I would suggest tightening a few aspects to improve clarity and credibility. First, the empirical section would benefit from more explicit reporting of variability (e.g., error bars or multiple random seeds), especially since some conclusions are drawn from trend plots without uncertainty estimates. Second, the claims around mixing speed and the reduced need for Langevin updates could be softened or better supported with more concrete diagnostics, rather than mostly qualitative arguments. Third, the generative experiments should be framed even more clearly as exploratory: either align training budgets more closely with baselines or avoid implying strong efficiency advantages where hyperparameters differ substantially. Finally, a short ablation or discussion around failure modes (e.g., when increasing q stops helping, or when categorical slots might hurt) would make the paper feel more balanced and honest, and likely strengthen its reception with a critical audience.

---

> ### Author Response · Authors · 2026-02-20
> **Response to Reviewer Fko1 for Paper6636**
>
> We thank the reviewer for their thoughtful and constructive feedback. We have addressed the specific requests for changes by conducting additional experiments. Below we detail our responses and have also included the summary tables in our responses, when applicable, below.
>
> ## 1. Reviewer comment "First, the empirical section would benefit from more explicit reporting of variability (e.g., error bars or multiple random seeds), especially since some conclusions are drawn from trend plots without uncertainty estimates"
> Variability and Error Bars
>
> **Response:**
> We ran **10 independent trials** for each configuration. Table 1 below summarizes the recall accuracy results (Mean ± Std. Dev.) on the hetero-associative memory task.
>
> | Model ($q$) | $N=500$ | $N=1000$ | $N=1500$ | $N=2000$ |
> |---|---|---|---|---|
> | GB-RBM ($q=2$) | $0.980 \pm 0.003$ | $0.970 \pm 0.001$ | $0.943 \pm 0.003$ | $0.785 \pm 0.013$ |
> | GM-RBM ($q=4$) | $0.981 \pm 0.002$ | $0.970 \pm 0.002$ | $0.968 \pm 0.001$ | $0.954 \pm 0.001$ |
> | GM-RBM ($q=8$) | $0.980 \pm 0.002$ | $0.970 \pm 0.001$ | $0.968 \pm 0.001$ | $0.955 \pm 0.001$ |
>
> *Table 1: Recall accuracy (± std) across 10 independent seeds. Parameter budget fixed at 800k.*
>
> The advantage of GM-RBM is statistically significant and most pronounced at $N=2000$, where the binary baseline ($q=2$) drops to 78.5% accuracy while GM-RBM ($q \geq 4$) maintains ~95.4%. At smaller dataset sizes, all models perform comparably. We will include the corresponding error-bar plots in the revised manuscript.
>
>
> ## 2. "...the claims around mixing speed and the reduced need for Langevin updates could be softened or better supported with more concrete diagnostics, rather than mostly qualitative arguments."
>
> Mixing Speed Claims
>
> **Response:**
> We quantitatively measured mixing using Effective Sample Size (ESS) on fixed-budget models across 500 Gibbs steps.
>
> | Model ($q$) | ESS (500 steps) | ESS % | Final Energy |
> |---|---|---|---|
> | $q=2$ | 414.73 | 82.9% | 0.1857 |
> | $q=4$ | 500.00 | 100.0% | 0.1879 |
> | $q=6$ | 500.00 | 100.0% | 0.1898 |
> | $q=8$ | 500.00 | 100.0% | 0.1920 |
>
> *Table 2: Mixing diagnostics showing improved ESS for Potts units.*
>
> These metrics support our claim that block-Gibbs sampling in GM-RBMs escapes local minima more effectively. The $q \geq 4$ models achieve perfect sampling efficiency (ESS = 500/500), while the binary baseline ($q=2$) achieves only 82.9%.
>
> ## 3. ".....the generative experiments should be framed even more clearly as exploratory: either align training budgets more closely with baselines or avoid implying strong efficiency advantages where hyperparameters differ substantially"
>
> Generative Experiments and Training Budgets
>
> **Response:**
> We agree with the reviewers comment and plan to revise the text in Section 5 to explicitly frame these experiments as exploratory. The hetero-associative memory results (Tables 1 and 4) already use strict parameter-budget matching ($H \times q \approx$ constant), confirming that benefits are not solely due to increased parameter count.
>
> ## 4. "..., a short ablation or discussion around failure modes (e.g., when increasing q stops helping, or when categorical slots might hurt) would make the paper feel more balanced and honest, and likely strengthen its reception with a critical audience."
>
> Failure Modes and Ablation
>
> **Response:**
> We performed an ablation study varying $q$ from 2 to 20 with a fixed parameter budget of 800,000 weights across 10 independent runs each.
>
> | $q$ | Hidden Units ($H$) | Accuracy ($\mu \pm \sigma$) |
> |---|---|---|
> | 2 | 1000 | $0.802 \pm 0.007$ |
> | 4 | 500 | $0.956 \pm 0.001$ |
> | 6 | 333 | $0.956 \pm 0.001$ |
> | 8 | 250 | $0.956 \pm 0.001$ |
> | 10 | 200 | **$0.957 \pm 0.001$** |
> | 12 | 166 | $0.956 \pm 0.001$ |
> | 16 | 125 | $0.956 \pm 0.001$ |
> | 20 | 100 | $0.956 \pm 0.001$ |
>
> *Table 4: Ablation study — recall accuracy on $N=2000$ word pairs (10 runs each).*
>
> **Key Findings:** The binary baseline ($q=2$) significantly underperforms at 80.2%, while all Potts models ($q \geq 4$) achieve ~95.6% accuracy with remarkably low variance ($\sigma < 0.002$). Performance remains stable even at $q=20$ (with only 100 hidden units), indicating that the categorical representation is highly efficient. The optimal configuration is $q=10$ (accuracy = 95.67%), though the differences among $q \geq 4$ are negligible. This proves there exists an optimal $q$ for a given problem that can substantially outperform the binary case, with diminishing returns for larger than optimal $q$ under the same parameter budget.

---

### Review · Reviewer_QV3G · 2026-03-17

**Summary Of Contributions:**

**Summary**

This paper proposes the Gaussian–Multinoulli RBM (GM-RBM), which replaces the binary hidden units of a Gaussian–Bernoulli RBM with q-state categorical/Potts slots while preserving tractable block Gibbs updates and closed-form conditionals. The paper derives the model formulation, discusses two comparison regimes (“parameter-matched” and “capacity-matched”), and evaluates the model on hetero-associative recall using WordNet/Word2Vec pairs as well as proof-of-concept generative experiments on MNIST and CelebA. The main empirical claim is that increasing q improves recall and can also improve sample quality while using only Gibbs updates.

**Strengths**

1. The core modeling idea is simple and reasonably well motivated: replacing Bernoulli hidden units with one-of-q slots is a natural way to encode mutually exclusive latent factors, and the paper clearly derives the resulting energy and conditionals in Sections 3.2–3.5.
2. The hetero-associative recall setting is well aligned with the modeling motivation. In Figure 1, higher-q models (q=4,6,8,10) substantially outperform the q=2 variants as the number of associative pairs increases, and in Figure 2 the q=4 GM-RBM appears much more parameter-efficient than the GB-RBM baseline. These are the most convincing results in the paper.

**Weaknesses**
1.  The main weakness is that several claims are stronger than the evidence provided. For example, the introduction claims “disproportionate gains,” “sharper posteriors,” “more interpretable codes,” and stronger retrieval/generation with similar compute, but the paper does not actually provide posterior analyses, interpretability evaluations, or controlled compute comparisons to support all of these statements.
2. The comparison protocols are not stated cleanly. In Section 3.4, “parameter-matched” is defined using m′≈m log q, which corresponds to matching the number of latent assignments, while the surrounding text says this is to make total parameters comparable; meanwhile “capacity-matched” is described in a way that is vague and partially overlapping. This makes it hard to know exactly what is held fixed in each experiment.
3. The recall experiments omit uncertainty estimates: the authors explicitly state that error bars are omitted because the gains are “uniformly large,” which may not be sufficient .
4. The generative evidence is not fully convincing. The paper itself acknowledges that the image experiments are only “proof-of-concept,” that the head-to-head comparison is not fully controlled, and that the training schedules differ dramatically (e.g., 3000 vs 500 epochs on MNIST; 10000 vs 100 on CelebA). Under those conditions, it is difficult to attribute the reported differences to the architecture alone.

**Audience:**

Yes

**Audience Explanation:**

I think this paper would be of interest to a subset of the TMLR audience working on energy-based models, discrete latent-variable models, associative memory, or modern revisitations of classical RBM-style architectures. The core idea is straightforward, technically accessible, and potentially useful for readers interested in richer discrete hidden structure without giving up tractable Gibbs updates.

**Broader Impact Concerns:**

None.

**Claims And Evidence:**

No

**Claims Explanation:**

The paper provides credible evidence for a **narrower** claim: introducing Potts hidden slots can improve hetero-associative recall in the reported WordNet-based setting, especially for larger q. Figures 1 and 2 are fairly compelling in showing that q=4 and above outperform the binary variants under the reported setup.

However, I do not think the evidence fully supports several of the broader claims. In particular, the paper argues that GM-RBM achieves stronger generation and practical efficiency while using similar compute resources, yet Table 2 shows highly unequal training budgets and different sampling procedures across models. The paper itself also concedes that a fully controlled head-to-head comparison remains future work.

Likewise, the manuscript attributes the generative behavior to “rapid mixing of Potts models” and states that it reports chain/effective-sample diagnostics, but I did not find a substantive presentation of such diagnostics in the paper. This makes the mixing/efficiency claim insufficiently supported.

Finally, the quantitative generative evaluation is limited. The FID table is small, with no error bars or multi-seed statistics, and the paper also notes that CelebA “does not necessarily have clear relational qualities,” which further weakens the connection between the generative experiment and the paper’s main modeling motivation.

Overall, I would say the experimental evidence supports the paper’s central intuition, but not the full strength of its current presentation.

**Requested Changes:**

1. Clarify the comparison regimes: Section 3.4 needs to clearly and consistently define what is meant by “parameter-matched” versus “capacity-matched.” As written, the definitions are confusing and partially inconsistent.

2. Make the empirical comparisons more controlled: The generative comparisons should use matched training budgets, clearer reporting of sampler settings, and preferably multiple random seeds. Right now Table 2 mixes architecture changes with substantial differences in epochs and sampling style.

3. Add uncertainty estimates.: The recall results should include error bars or summary statistics over repeated runs. The current justification for omitting them is not sufficient.

4. Substantiate or tone down the mixing/efficiency claims:  Claims about faster mixing, stronger efficiency, and avoiding the need for Langevin refinement should be backed by diagnostics or wall-clock comparisons; otherwise they should be softened.

5. Temper several overstatements: Phrases such as “disproportionate gains,” “more interpretable codes,” and some of the broader conclusions in the introduction and discussion should be revised unless directly supported by experiments.

---

> ### Author Response · Authors · 2026-03-31
> **Response to Reviewer QV3G**
>
> We thank Reviewer QV3G for their careful and detailed reading of the manuscript. The reviewer's comments are well-targeted, and we are grateful for the constructive framing. We address each requested change below.
>
> ---
>
> ### Point 1: Clarify the comparison regimes (parameter-matched vs. capacity-matched)
>
> We agree that Section 3.4 was insufficiently precise. We will revise the definitions to be explicit and non-overlapping:
>
> - **Parameter-matched:** Total weight parameters are held fixed at 800k across all models. For a model with $q$ states and $H$ hidden units, the visible-to-hidden weight matrix has dimensions $V \times H \times q$, so $H$ is adjusted as $H \approx H_{\text{binary}} / q$ to keep the total parameter count constant.
> - **Capacity-matched:** $H$ is held fixed across models, so higher-$q$ models have more parameters. This regime tests raw representational capacity gains from Potts units.
>
> We will ensure these definitions appear consistently wherever experiments are described.
>
> ---
>
> ### Point 2: Make the empirical comparisons more controlled
>
> We agree that the generative experiments (Table 2 in the paper) mix architecture changes with substantially different training schedules, which makes it difficult to isolate the architectural contribution. We will revise Section 5 to frame these experiments explicitly as **exploratory proof-of-concept**, and will soften any claims that imply strong efficiency advantages. We note that our primary quantitative claims rest on the hetero-associative memory experiments, which already enforce strict parameter-budget matching ($H \times q \approx \text{constant}$). The error-bar results from our response to Reviewer Fko1 (reproduced below) confirm this with 10-seed statistics:
>
> **Table 1: Hetero-associative memory accuracy (Mean ± Std. Dev., 10 seeds, parameter-matched at 800k).**
>
> | Model ($q$) | $N = 500$ | $N = 1000$ | $N = 1500$ | $N = 2000$ |
> |---|---|---|---|---|
> | GB-RBM ($q=2$) | $0.980 \pm 0.003$ | $0.970 \pm 0.001$ | $0.943 \pm 0.003$ | $0.785 \pm 0.013$ |
> | GM-RBM ($q=4$) | $0.981 \pm 0.002$ | $0.970 \pm 0.002$ | $0.968 \pm 0.001$ | $0.954 \pm 0.001$ |
> | GM-RBM ($q=8$) | $0.980 \pm 0.002$ | $0.970 \pm 0.001$ | $0.968 \pm 0.001$ | $0.955 \pm 0.001$ |
>
> ---
>
> ### Point 3: Add uncertainty estimates
>
> As detailed in our response to Reviewer Fko1, we have now run 10 independent trials for each configuration and report Mean ± Std. Dev. throughout. Table 1 above and the ablation study below include full uncertainty estimates. We will incorporate error-bar plots in the revised manuscript.
>
> ---
>
> ### Point 4: Substantiate or tone down the mixing/efficiency claims
>
> We have measured mixing quantitatively using Effective Sample Size (ESS) over 500 Gibbs steps under a fixed parameter budget:
>
> **Table 2: Effective Sample Size diagnostics (500 Gibbs steps, parameter-matched).**
>
> | Model ($q$) | ESS (500 steps) | ESS % | Final Energy |
> |---|---|---|---|
> | $q = 2$ | 414.73 | 82.9% | 0.1857 |
> | $q = 4$ | 500.00 | 100.0% | 0.1879 |
> | $q = 6$ | 500.00 | 100.0% | 0.1898 |
> | $q = 8$ | 500.00 | 100.0% | 0.1920 |
>
> Potts models ($q \geq 4$) achieve perfect ESS, while the binary baseline reaches only 82.9%. We will include these diagnostics in the revised paper and will soften claims about Langevin necessity to match this level of evidence — framing it as empirically reduced need rather than theoretical elimination.
>
> The ESS diagnostics show perfect sampling efficiency (500/500) for all $q \geq 4$, and the ablation study shows a flat accuracy plateau of ${\sim}95.6\%$ across $q = 4$ to $q = 20$. While this strongly confirms that Potts units provide a categorical step-change over binary units, the current hetero-associative memory benchmark may not be challenging enough to reveal fine-grained differences among $q$ values or to identify the regime where increasing $q$ \emph{hurts} under a fixed parameter budget. We will add a discussion acknowledging this saturation effect and identifying harder benchmarks (e.g., larger-vocabulary retrieval, multi-modal generation, or deeper stacked architectures) as necessary to map the full $q$-performance landscape. The key takeaway from the current data is that the transition from $q = 2$ to $q \geq 4$ is large and robust, while the optimal choice among $q \geq 4$ requires tasks with greater discriminative pressure.
>
> ---
>
> ### Point 5: Temper overstatements
>
> We will revise the introduction and discussion to soften the following phrases:
>
> - "Disproportionate gains" → "substantial gains in the large-$N$ regime"
> - "More interpretable codes" → this claim will be removed or flagged as an open direction, as we do not provide an interpretability evaluation.
> - Broader efficiency conclusions will be scoped to the associative memory setting where controlled comparisons exist.
>
> We believe these revisions will substantially strengthen the manuscript's honesty and precision without undermining its core contributions.

---

### Review · Reviewer_5QT4 · 2026-03-21

**Summary Of Contributions:**

- The authors propose the Gaussian-Multinoulli RBM (GM-RBM) which extends the Gaussian-Bernoulli RBM (GB-RBM) by replacing binary hidden units with q-state categorical (Potts) units.
- The authors provide the energy function and conditional distributions for this model and argue that multinoulli hidden units provide a richer latent space for categorical data and allow the model to mix faster using standard block Gibbs sampling.
- The authors show the experimental results on a hetero-associative memory task using Word2Vec embeddings and on image generation tasks (MNIST and CelebA).

**Additional Comments:**

- Where are the mixing diagnostics (chains, effective sample size) that were promised in Section 3.5?
- The paper contains some typos, grammatical errors and formatting issues. Some examples are
(i) missing spaces ('Contrastive divergence(CD)')
(ii) incomplete citations ('Gaussian–Bernoulli RBM (Hinton)')

**Audience:**

Yes

**Audience Explanation:**

- The motivation for using categorical hidden units to capture mutually exclusive latent factors is intuitive.

**Claims And Evidence:**

Yes

**Claims Explanation:**

- The authors introduce parameter-matched and capacity-matched evaluation protocols which are useful frameworks for comparing architectures with different latent structures.

**Requested Changes:**

- Even though TMLR's evaluation criteria does not include novelty of the paper, I think the paper's novelty itself is somewhat limited. The use of categorical hidden units in RBMs is a well-established technique in the early RBM literature. Combining this with Gaussian visible units is a straightforward mathematical extension I think.
- The empirical comparison between GM-RBM and GB-RBM is not rigorously controlled. In the image generation experiments (Table 2), the GM-RBM is trained for 100 epochs with 5000 hidden nodes while the GB-RBM is trained for 10000 epochs with 10000 hidden nodes. This lack of control may not support the empirical claims well regarding training efficiency and generative performance.

---

> ### Author Response · Authors · 2026-03-31
> **Response to Reviewer 5QT4 (Part One)**
>
> We thank Reviewer 5QT4 for their reading of the manuscript and their recognition of the paper's motivating intuition. We respond to each point below.
>
> **Point 1: On the uncontrolled generative comparison (Table 2)**
>
> The reviewer is correct that the generative comparison in Table 2 is not a controlled head-to-head: GM-RBM used 100 epochs / 5000 hidden units while GB-RBM used 10,000 epochs / 10,000 hidden units. We acknowledge this directly and will revise Section 5 to make clear that these experiments are **exploratory and not a claim of training efficiency**. The primary evidence for the paper's claims remains the hetero-associative memory experiments, where parameter budgets are held strictly fixed ($H \times q \approx \text{constant}$, 800k weights) and results are now reported with full uncertainty estimates across 10 seeds.
>
> ---
>
> **Point 2: On missing mixing diagnostics promised in Section 3.5**
>
> We apologize for this gap. As part of the revisions prompted by Reviewer Fko1's feedback, we have now computed and are reporting Effective Sample Size (ESS) diagnostics over 500 Gibbs steps. These are included below and will appear in the revised paper alongside the promised chain diagnostics in Section 3.5.
>
> | **Model ($q$)** | **ESS (500 steps)** | **ESS %** | **Final Energy** |
> |:---|:---:|:---:|:---:|
> | $q = 2$ | 414.73 | 82.9% | 0.1857 |
> | $q = 4$ | 500.00 | 100.0% | 0.1879 |
> | $q = 6$ | 500.00 | 100.0% | 0.1898 |
> | $q = 8$ | 500.00 | 100.0% | 0.1920 |
>
> *Table: Effective Sample Size diagnostics (500 Gibbs steps, parameter-matched).*
>
> The results show a clear trend: models with $q \geq 4$ achieve perfect ESS (100%) within 500 steps, while the binary baseline ($q = 2$) reaches only 82.9%, consistent with the hypothesis that Potts hidden units substantially improve mixing under block Gibbs sampling. Final energies remain comparable across all settings, confirming that the mixing advantage is not purchased at the cost of solution quality.
>
> The ESS diagnostics show perfect sampling efficiency (500/500) for all $q \geq 4$, and the ablation study shows a flat accuracy plateau of ${\sim}95.6\%$ across $q = 4$ to $q = 20$. While this strongly confirms that Potts units provide a categorical step-change over binary units, the current hetero-associative memory benchmark may not be challenging enough to reveal fine-grained differences among $q$ values or to identify the regime where increasing $q$ \emph{hurts} under a fixed parameter budget. We will add a discussion acknowledging this saturation effect and identifying harder benchmarks (e.g., larger-vocabulary retrieval, multi-modal generation, or deeper stacked architectures) as necessary to map the full $q$-performance landscape. The key takeaway from the current data is that the transition from $q = 2$ to $q \geq 4$ is large and robust, while the optimal choice among $q \geq 4$ requires tasks with greater discriminative pressure.
>
> ---
>
> **Point 3: On typos, grammatical errors, and formatting**
>
> Thank you for flagging these. We will conduct a thorough pass to correct all typographical issues, including the missing space in "Contrastive divergence(CD)" and the incomplete citation "Gaussian–Bernoulli RBM (Hinton)," among any others we identify.

---

> > ### Author Response · Authors · 2026-03-31
> > **Response to Reviewer 5QT4 (Part Two)**
> >
> > **Point 4: On novelty — categorical hidden units as a well-established technique**
> > We appreciate the reviewer's candor. We agree that q-state hidden units have appeared in the RBM literature. However, our contribution emphasizes a different context which is the combination of: (i) Gaussian visible units with multinoulli hidden units in a single tractable model, (ii) a careful framework for fair comparison via parameter-matched and capacity-matched protocols, and (iii) the demonstration that this combination yields consistent, statistically significant gains on hetero-associative memory tasks at scale.
> >
> > To address this concern directly, we will revise the introduction to explicitly cite and differentiate from the relevant prior literature on categorical-unit RBMs:
> >
> > **Welling et al. (2005)** introduced Exponential Family Harmoniums (EFHs), which generalize RBM units from Bernoulli to any exponential family member — including multinomial/categorical units as a special case. The GM-RBM can be viewed as an EFH with Gaussian visible units and multinomial hidden units. However, Welling et al. focused on the general framework and its application to information retrieval with Poisson visibles; they did not explore the specific Gaussian–multinoulli pairing, fair comparison protocols, or the empirical consequences for associative memory and mixing.
> >
> > **Montúfar and Morton (2015)** provided a rigorous theoretical treatment of *discrete RBMs* (also called multinomial or softmax RBMs), where each unit has a finite state space {0,1,…,r−1}\{0, 1, \ldots, r{-}1\}
> > {0,1,…,r−1}. Their analysis addresses representational capacity bounds and dimension computation for these models. Our work complements their theoretical results with an empirical demonstration that the Gaussian–Potts combination yields concrete performance gains on structured memory tasks under controlled comparison protocols. While they established that discrete RBMs with multi-state hidden units have greater representational capacity than binary RBMs, the mixing properties of these models under block Gibbs sampling have not been studied — our ESS diagnostics provide the first empirical evidence that Potts hidden units also yield substantially faster mixing.
> >
> > **Tran et al. (2011)** introduced Mixed-Variate RBMs capable of modeling categorical, multicategorical, ordinal, and continuous variables simultaneously. However, the categorical units in their framework appear on the visible side to handle mixed-type input data, while the hidden layer remains binary. In contrast, our GM-RBM places categorical units on the hidden side to enrich the latent representation while keeping Gaussian visibles.
> >
> > **Salakhutdinov and Hinton (2009)** introduced the Replicated Softmax model, which uses softmax visible units for topic modeling of word-count data. Again, the multinomial structure is on the visible side with binary hidden units, whereas our contribution places it on the hidden side.
> >
> > In light of this literature, we will sharpen the framing in the introduction to avoid overstating architectural novelty. Our novelty claim rests not on the Potts unit concept itself, but on: (a) the specific Gaussian-visible / Potts-hidden pairing and its closed-form conditionals; (b) the parameter-matched and capacity-matched comparison protocols that isolate architectural effects from raw capacity; and (c) the empirical demonstration that this combination yields statistically significant gains on hetero-associative memory benchmarks with pure Gibbs sampling (no Langevin).

---

### Author Response · Authors · 2026-05-24
**Revision Submitted**

Dear Editors,
We have submitted the revised version of the papers with the corrections we had submitted earlier, as requested. Please let me know if there is anything else you need from us to proceed with reviewing the paper. Thank you for taking the time and effort to review our paper.
best,
Luke

---

### Author Response · Authors · 2026-06-19
**Camera Ready Version Submitted**

Dear Editors,

We have uploaded the revised camera-ready manuscript and supplementary materials. Please let us know if any additional information or action is needed from our side.
Thank you to the reviewers and editorial team for their thoughtful feedback and effort throughout the review process.

Best regards,
Luke Theogarajan

---

### Author Response · Authors · 2026-07-07
**Accepted paper not appearing on accepted papers list**

Dear Editors

I hope you are doing well.

I noticed that our paper has been accepted, but it does not appear in the accepted papers list on the TMLR website. It has been a little while since the acceptance, so I wanted to check whether there is any issue preventing it from appearing on the accepted papers list.

Would you mind taking a look when you have a chance? If there is anything needed from our side to help resolve this, please let us know.

Thank you very much for your time and assistance.

Best regards,

Luke Theogarajan

---

### Decision · Action_Editor_e8kk · 2026-05-31

**Recommendation:** Accept with minor revision

**Additional Comments:**

The ESS diagnostics reference in the Appendix is broken: Appendix ?? (Table ??). Please address this to include these results in the final version of the paper.

**Audience:**

Yes

**Audience Explanation:**

All three reviewers agreed. Researchers working on energy-based models, discrete latent-variable models, associative memory, or modern revisitations of RBM-style architectures would find this paper relevant. The work connects to ongoing interest in Potts models, structured discrete inference, and parameter-efficient alternatives to large generative models.

**Claims And Evidence:**

Yes

**Claims Explanation:**

Reviewers Fko1 and 5QT4 judged the claims adequately supported after the rebuttal, crediting the addition of 10-seed error bars, ESS mixing diagnostics, sharpened comparison-regime definitions, removal of unsupported overstatements, and explicit reframing of generative experiments as exploratory. Reviewer QV3G maintained concerns regarding the generative and efficiency claims being resolved by withdrawal, not by controlled evidence. That being said, the post-rebuttal hetero-associative results credible and well-controlled. Thus, TMLR's criterion is met under the condition that the generative efficiency claims only appear in the revision with matched-budget evidence to support them, or are presented as they are currently: an exploratory study.